# Virulent properties and genomic diversity of *Vibrio vulnificus* isolated from environment, human, diseased fish

Ampapan Naknaen,[1,2] Komwit Surachat,[3,4] Jutamas Manit,[2] Korakot Wichitsa-nguan Jetwanna,[5] Jumroensri Thawonsuwan,[6] Rattanaruji Pomwised[2]

**ABSTRACT** The incidence of *Vibrio vulnificus* infections, with high mortality rates in humans and aquatic animals, has escalated, highlighting a significant public health challenge. Currently, reliable markers to identify strains with high virulence potential are lacking, and the understanding of evolutionary drivers behind the emergence of pathogenic strains is limited. In this study, we analyzed the distribution of virulent genotypes and phenotypes to discern the infectious potential of *V. vulnificus* strains isolated from three distinct sources. Most isolates, traditionally classified as biotype 1, possessed the virulence-correlated gene-C type. Environmental isolates predominantly exhibited YJ-like alleles, while clinical and diseased fish isolates were significantly associated with the *nanA* gene and pathogenicity region XII. Hemolytic activity was primarily observed in the culture supernatants of clinical and diseased fish isolates. Genetic relationships, as determined by multiple-locus variable-number tandem repeat analysis, suggested that strains originating from the same source tended to cluster together. However, multilocus sequence typing revealed considerable genetic diversity across clusters and sources. A phylogenetic analysis using single nucleotide polymorphisms of diseased fish strains alongside publicly available genomes demonstrated a high degree of evolutionary relatedness within and across different isolation sources. Notably, our findings reveal no direct correlation between phylogenetic patterns, isolation sources, and virulence capabilities. This underscores the necessity for proactive risk management strategies to address pathogenic *V. vulnificus* strains emerging from environmental reservoirs.

**IMPORTANCE** As the global incidence of *Vibrio vulnificus* infections rises, impacting human health and marine aquacultures, understanding the pathogenicity of environmental strains remains critical yet underexplored. This study addresses this gap by evaluating the virulence potential and genetic relatedness of *V. vulnificus* strains, focusing on environmental origins. We conduct an extensive genotypic analysis and phenotypic assessment, including virulence testing in a wax moth model. Our findings aim to uncover genetic and evolutionary factors that drive pathogenic strain emergence in the environment. This research advances our ability to identify reliable virulence markers and understand the distribution of pathogenic strains, offering significant insights for public health and environmental risk management.

**KEYWORDS** vibriosis, genetic diversity, PRXII, fish pathogen, wax moth model

*Vibrio vulnificus* is a Gram-negative estuarine bacterium causing disease in both humans and aquatic animals (1). As a significant foodborne pathogen, it leads to primary septicemia, wound infections, and gastroenteritis, with a high fatality rate in humans. The incidence of *V. vulnificus* infections has increased in the subtropical regions of all continents, including in the USA, Europe, Singapore, and Thailand (2, 3). Beyond its

Address correspondence to Rattanaruji Pomwised, rattanaruji.p@psu.ac.th.

The authors declare no conflict of interest.

See the funding table on p. 15.

impact on humans, *V. vulnificus* is a notable aquatic pathogen causing vibriosis in aquatic cultures, including eel, tilapia, marine rainbow trout, and grouper, leading to economic losses (4–8). The pathogen enters the fish's body through gills and colonizes the intestine and anus, causing acute hemorrhagic septicemia that can be fatal (4, 9).

To distinguish between virulent and non-virulent strains, genetic elements, geographic range, and pathogenicity of *V. vulnificus* have been investigated. *V. vulnificus* is traditionally classified into biotypes 1, 2, and 3 based on biochemical properties, biotypes 1 and 2 predominating in human infections and eel pathogenicity, respectively. Biotype 3, a hybrid of biotypes 1 and 2, is found in restricted areas (10). Advancements in sequencing technologies have enabled the classification of *V. vulnificus* into five phylogenomic lineages (L1–L5) through whole-genome analysis (11). Most clinical and environmental isolates (biotypes 1 and 2) belong to lineages L1 and L2, while fish pathogens (biotype 3) are typically grouped in L3. Some biotype 1 strains fall into L4 and L5, which are associated with specific geographical areas.

The diversity of *V. vulnificus* is further elucidated through multiple-locus variable-number tandem repeat analysis (MLVA), which assists in constructing phylogenetic trees and epidemiological data (12, 13). While harboring a range of virulent elements, human and fish isolates can be found across various phylogenetic groups (14). Based on these technologies, the evolution of *V. vulnificus* and relationships among isolated sources that underlie the emergence of virulent strains are gradually revealed.

Pathogenic *V. vulnificus* strains are often identified by the presence of putative virulent factors and associated genes, such as the virulence-correlated gene (vcg), cell wall elements, cytotoxicity systems, and mechanisms for cell attachment, adhesion, and motility (10). Traditionally, *V. vulnificus* has been classified based on these genetic markers to distinguish clinical/pathogenic strains from environmental ones (15). Despite its recognized impact, there remains a limited understanding of the virulence mechanisms and genetic traits that drive the pathogenicity of *V. vulnificus*, particularly in strains originating from environmental sources. This study aims to delineate the pathogenic potential of clinical, diseased fish, and environmental *V. vulnificus* isolates through a comprehensive analysis of their genotype and phenotype traits. We seek to offer new insights into their virulence capabilities and evolutionary drivers, thereby contributing to risk assessment strategies crucial for public health and aquaculture management.

## RESULTS

### Assessment of potential virulent *V. vulnificus* strains by detection of virulence genes

To investigate tentative virulent strains from different isolation sources by the presence of virulence genes, 85 *V. vulnificus* isolates used in this study were categorized into three groups based on their original sources: environmental isolates [$n = 41$ (48%)], clinical isolates [$n = 10$ (12%)], and diseased fish isolates [$n = 33$ (39%)]. Prior to virulent gene region detection, traditional classification based on biochemical properties was used to determine *V. vulnificus* to biotypes 1, 2, and 3 (16, 17). Biotype 1 has been abundant in various environmental and clinical settings, whereas most fish pathogens fall into biotype 2 (4). Here, most of *V. vulnificus* (96%) belonged to biotype 1 (indole producing), while two environmental isolates and a diseased fish isolate fell into biotype 2. The traditional classification for *V. vulnificus* relies on the *vcg* associated with isolate source and pathogenicity (15). The *vcg-C* and *vcg-E* have been predominant in clinical and environmental strains, respectively. This study demonstrated that all *V. vulnificus* isolates were classified as vcg-C type. The prevalence of *vcg-C* has been highly detected in warm water temperature habitats (18–20), while vcg-E type is rarely found in environmental niches from the tropical climate region. Reports from China (21) and Japan (22) show that most environment isolates are vcg-C positive. *V. vulnificus* isolated from shrimp (23) and fish (5) in Thailand was also classified into *vcg-C* type, implying that *vcg-C* type in estuarine environments might be associated with aquaculture infections. The predominance of vcg-C in warm water environments suggests a potential shift in virulence traits.

However, whole-genome analysis reveals that *vcg-C* type in *V. vulnificus* strains does not necessarily correlate with their source of isolation, be it clinical or environmental (24). Therefore, classification based on *vcg* type cannot forecast the virulence potential.

To further determine whether *V. vulnificus* harbored other associated virulent genes, we detected four virulent-related regions, including *manIIA*, *nanA*, CM-like allele/YJ-like allele, and pathogenicity region XII (PRXII) (Fig. 1).

   i. The *manIIA* was predominantly found in the 80 isolates (94%), which also showed the ability to ferment mannitol (Fig. 1). Previous studies suggested that clinical strains often possess the *manIIA* gene, associated with mannitol fermentation, as

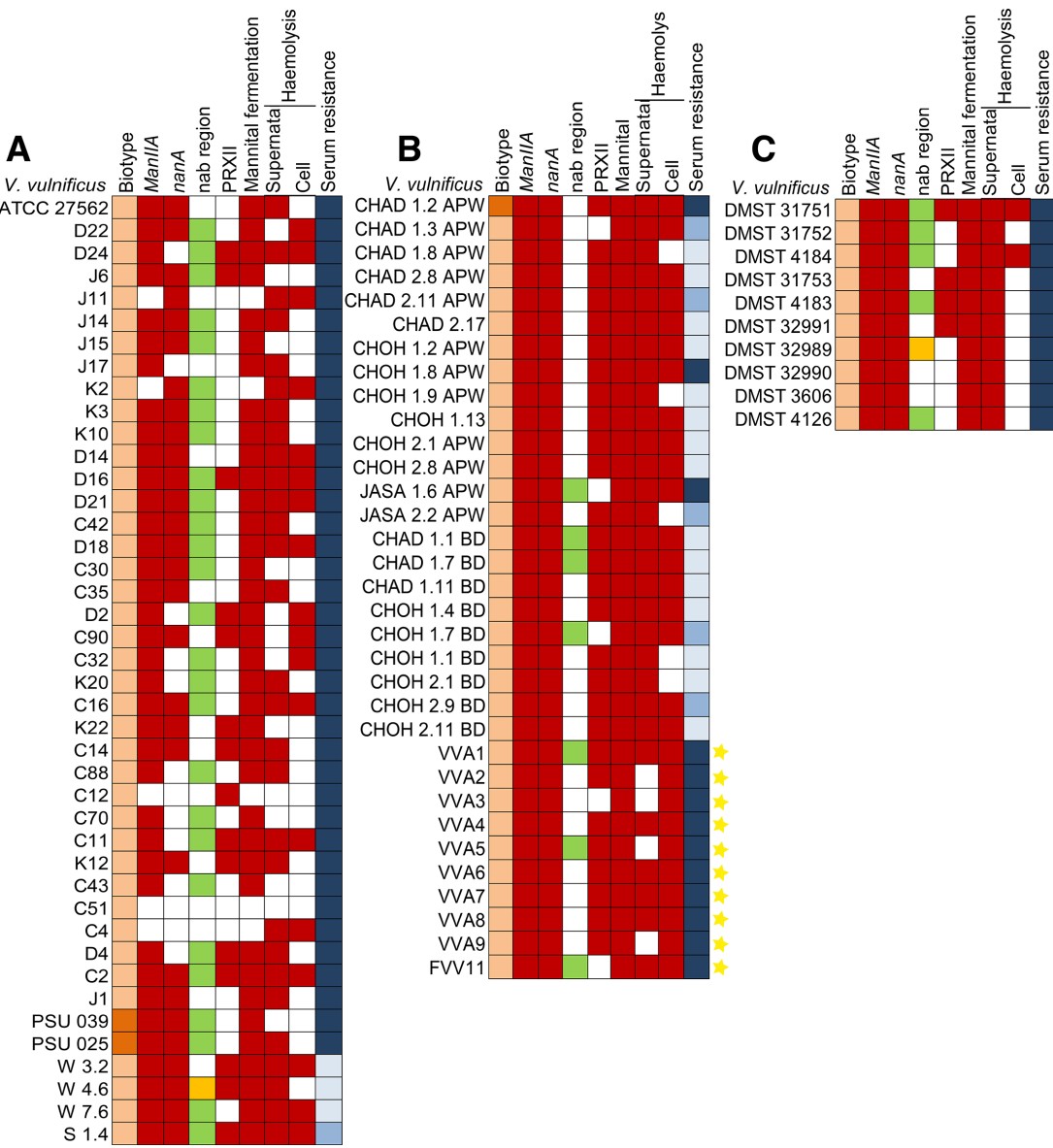

**FIG 1** Eighty-five *V. vulnificus* strains revealed the existence of divergent genetic elements and phenotypic characteristics among isolated sources, including (A) environmental isolates, (B) diseased fish isolates, and (C) clinical isolates. *V. vulnificus* isolates were classified into biotype 1 (light orange) and biotype 2 (dark orange). For the nab region, green and yellow shades represent YJ-like and CM-like alleles, respectively. The serum sensitivity was categorized into three levels, including resistant (navy blue), intermediate (denim blue), and sensitive (sky blue). The red color represents the presence of the targeting genes. The yellow star indicates the fish pathogen.

well as genes involved in sialic acid catabolism (*nanA, nanE,* and *nanK*) and the type IV secretory pathway (25).

ii. The catabolic sialic acid pathway involves three enzymes, lyase/aldolase (*nanA*), kinase (*nanK*), and epimerase (*nanE*) (26). These enzymes contribute to the decoration of the lipopolysaccharide, aiding the bacterium in evading the immune system (27). Our results showed that clinical isolates (Fig. 1D) and diseased fish isolates (Fig. 1B and C) have an overall *nanA* gene profile, compared with environmental isolates (68%) (Fig. 1A). Previous studies reported that clinical strains significantly carried *nanA* more than environmental strains (26, 28).

iii. The nab region, essential for nonulosonic acid (NulOs) biosynthesis, is involved in biofilm formation, autoagglutination, and motility (29). The allele types of nap region were designated via two reference genomes, CMCP6 (biotype 1, human pathogen) as CM-like allele and YJ016 (biotype 1, fish pathogen) as YJ-like allele (30). The nab region shows divergence in allele types (CM-like and YJ-like) between clinical and environmental strains (29). Contrary to previous reports, our study found significant presence of YJ-like alleles in 66% of environmental isolates ($P = 0.049$), while only one clinical isolate (DMST 32990) and one diseased fish isolate (W 4.6) were positive for the CM-like allele, as illustrated in Fig. 1. This indicates a more complex relationship between nab allele types and strain sources than previously understood.

iv. The genomic island XII region (PRXII), located on the small chromosome (VVA1613 to VVA1636), harbors two putative chondroitinase genes, an ABC transport system, the putative arylsulfatase A gene cluster, and putative methyl-accepting chemotaxis protein (31). This region is associated with high-virulence *vcg-C* strains and plays a role in the bacterium's adaptability to both environment and human hosts (25, 31). Several studies report that the presence of PRXII in virulent strains was significantly found in clinical isolates than in environmental isolates (29). However, whole-genome comparisons indicate that PRXII can be present in both clinical and environmental isolates (24). Our result showed that PRXII was found significantly more in diseased fish isolates (85%; Fig. 1B and C) compared to environmental isolates (37%; $P = 0.001$, Fig. 1A) or clinical isolates (40%; $P = 0.012$, Fig. 1D).

## Hemolytic activity and serum resistance

Hemolytic activity has been previously shown by various virulent strains from diseased fish and clinical isolates (29, 32). Here, the hemolytic activity of culture supernatant was significantly more prevalent in diseased fish isolates (88%; $P = 0.001$, Fig. 1B and C) and clinical isolates (100%; $P = 0.03$, Fig. 1D) compared to environmental isolates (68%; Fig. 1A). On the other hand, cell pellets from diseased fish isolates demonstrated hemolytic activity in 85% of cases (Fig. 1B and C). Compared to environmental isolates, diseased fish isolates exhibit intense hemolytic activity in both supernatants and cell pellets.

All clinical and environmental isolates showed the ability to resist human serum (Fig. 1A). The serum resistance appeared in 14 out of 37 diseased fish isolates (Fig. 1B and C). Resistance to human serum, a known virulence factor, is more commonly observed in *vcg-C* clinical isolates than in the environmental *vcg-E* type (33, 34). However, we found the serum resistance in all *vcg-C* clinical and *vcg-C* environmental isolates. This ability may be specific for *vcg-C* type regardless of the source of origin.

## Molecular typing of *V. vulnificus* strains based on MLVA and multilocus sequence typing (MLST)

Based on isolation sources, 20 *V. vulnificus* strains were selected to analyze the clonal groups by using MLVA. MLVA approach has been used as an epidemiological tool to distinguish bacterial populations with small genetic variations (35, 36). Eleven variable number tandem repeat (VNTR) loci were investigated in *V. vulnificus* isolates. The largest

variation in single sequence repeats (SSRs) was found in VVA-0375 (from 0 to 46 repeats). MLVA discriminated between 18 and 19 types, with genetic distances ranging from 70% to 85% among them. The discrimination power, evaluated using Simpson's diversity index (DI) revealed that VVA-0375 had the highest DI (0.984) with the highest number of tandem repeats (TR) (Fig. 2A). On the other hand, the lowest DI was observed in VVA-1615 (0.647), which spanned from 2 to 23 repeats (Fig. 2A). The clonal relationships among the isolates divided them into three main groups, including M1, M2, and M3

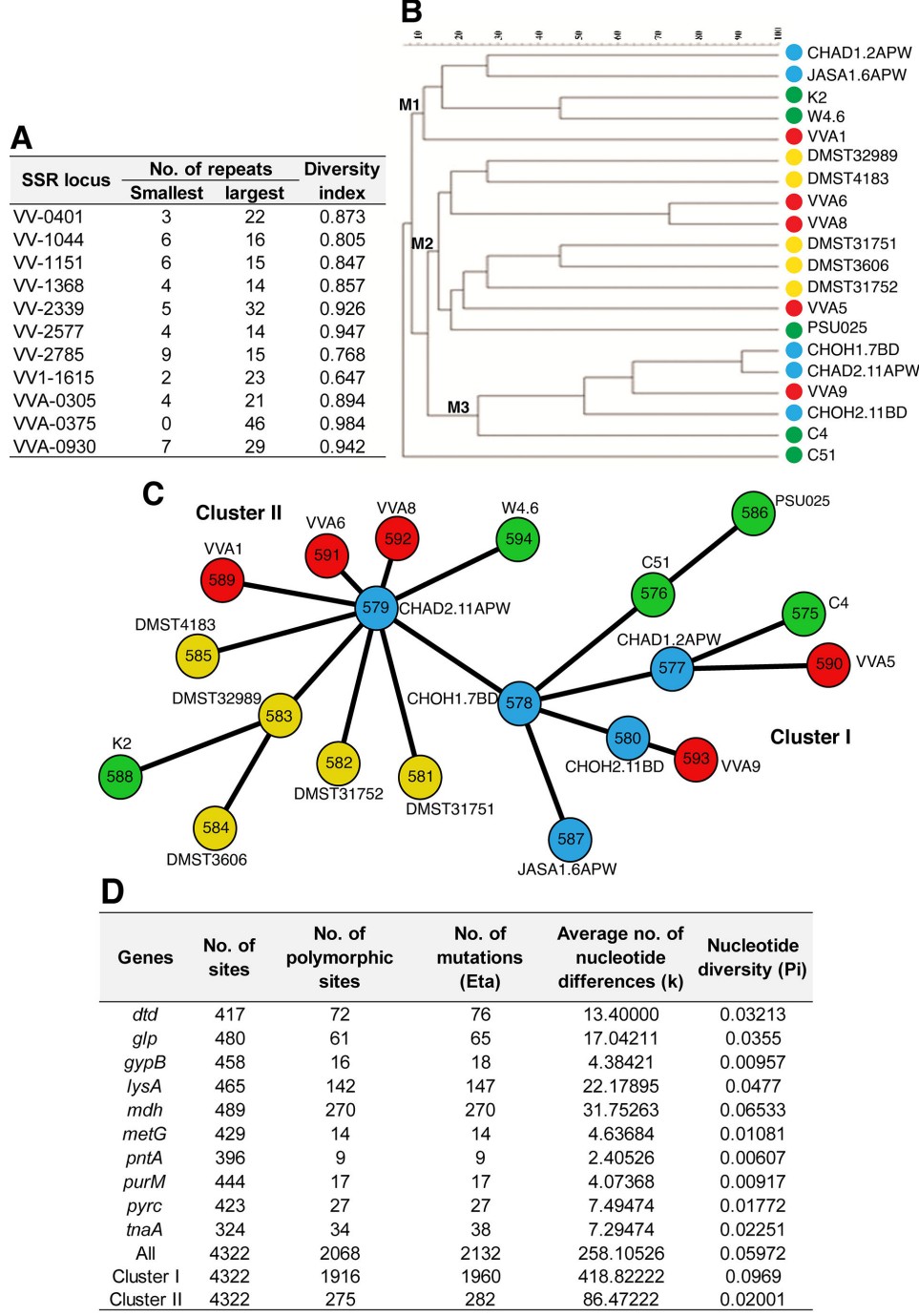

**A**

| SSR locus | No. of repeats | | Diversity index |
| | Smallest | largest | |
|---|---|---|---|
| VV-0401 | 3 | 22 | 0.873 |
| VV-1044 | 6 | 16 | 0.805 |
| VV-1151 | 6 | 15 | 0.847 |
| VV-1368 | 4 | 14 | 0.857 |
| VV-2339 | 5 | 32 | 0.926 |
| VV-2577 | 4 | 14 | 0.947 |
| VV-2785 | 9 | 15 | 0.768 |
| VV1-1615 | 2 | 23 | 0.647 |
| VVA-0305 | 4 | 21 | 0.894 |
| VVA-0375 | 0 | 46 | 0.984 |
| VVA-0930 | 7 | 29 | 0.942 |

**D**

| Genes | No. of sites | No. of polymorphic sites | No. of mutations (Eta) | Average no. of nucleotide differences (k) | Nucleotide diversity (Pi) |
|---|---|---|---|---|---|
| *dtd* | 417 | 72 | 76 | 13.40000 | 0.03213 |
| *glp* | 480 | 61 | 65 | 17.04211 | 0.0355 |
| *gypB* | 458 | 16 | 18 | 4.38421 | 0.00957 |
| *lysA* | 465 | 142 | 147 | 22.17895 | 0.0477 |
| *mdh* | 489 | 270 | 270 | 31.75263 | 0.06533 |
| *metG* | 429 | 14 | 14 | 4.63684 | 0.01081 |
| *pntA* | 396 | 9 | 9 | 2.40526 | 0.00607 |
| *purM* | 444 | 17 | 17 | 4.07368 | 0.00917 |
| *pyrc* | 423 | 27 | 27 | 7.49474 | 0.01772 |
| *tnaA* | 324 | 34 | 38 | 7.29474 | 0.02251 |
| All | 4322 | 2068 | 2132 | 258.10526 | 0.05972 |
| Cluster I | 4322 | 1916 | 1960 | 418.82222 | 0.0969 |
| Cluster II | 4322 | 275 | 282 | 86.47222 | 0.02001 |

The asterisk indicated that Tajima's D value was significantly deviated: *, P < 0.05, **, P < 0.01, and ***, P < 0.001.

**FIG 2** Genetic relatedness of selected *V. vulnificus* strains based on MLVA and MLST. (A) Characteristics of SSR loci used for MLVA. (B) Dendrogram of *V. vulnificus* strains based on MLVA profiles. (C) Dendrogram of *V. vulnificus* strains based on 10 housekeeping MLST genes. (D) Characteristics of the genes and clusters analyzed by MLST.

(Fig. 2B). Each group contained isolates from various sources. Five members in M1 came from three diseased fish isolates and two environmental isolates. M3 was composed of four diseased fish isolates and an environmental isolate. Five clinical isolates, three diseased fish isolates, and an environmental isolate constituted M2. Interestingly, one isolate showed uniqueness with 8% genetic distance. Members in M2 and M3 groups carried the PRXII more than did members in M1. Other virulence-correlated genes were equally distributed to all groups.

The 10 housekeeping MLST genes of the 20 isolates were successfully subtyped (37). A total of 163 new allelic types (ATs) were identified at *gypB*, *mdh*, *metG*, *dtdS*, *lysA*, *pyrC*, and *tnaA* using PubMLST (38), while 3 alleles of *glp*, 16 alleles of *pntA*, and 18 alleles of *purM* exactly matched with the ATs in the database (Table S3). The 20 novel distinct MLST sequence types (STs) represented by a single isolate were assigned as ST-575 to ST-594 based on the combination of ATs using PubMLST (Table S3; Fig. 2C). Recently, new STs of *V. vulnificus* isolated from diseased grouper in Thailand were reported as ST-595, ST-596, and ST-597 (39), indicating the high genetic diversity among *V. vulnificus*. Characteristics of the 10 genes were analyzed using the DnaSP program. The average nucleotide diversity (Pi) of the 10 loci was $0.02565 \pm 0.01945$, with the lowest degree of diversity observed for *pntA* (0.00607) (Fig. 2D). The *mdh* gene exhibited the highest nucleotide diversity with the greatest number of polymorphic sites ($n = 270$) and mutations ($n = 270$) among the strains. This was followed by *lysA*, which displayed the second highest diversity (Fig. 2D). The Tajima's D values of the nine genes were negative, indicating an excess of rare variants (40). The eight genes (*dtd*, *glp*, *gypB*, *metG*, *pntA*, *purM*, *pyrc*, and *tnaA*) were considered insignificant (Fig. 2D), suggesting that mutations on the DNA level do not affect fitness. However, the two significant Tajima's D values in *lysA* and *mdh* indicated a selective sweep and high frequency of rare variants. Only *metG* showed positive values for Tajima's D, suggesting this gene was under balancing selection.

To evaluate clonal relationships, a spanning tree was constructed based on the 10‑loci concatenated sequence of the 20 isolates. The tree, depicted in Fig. 2C, was divided into two divergent clusters, I and II. Nucleotide diversity and number of mutations of cluster I isolates (pi = 0.0969; Eta = 1960) were significantly higher, by approximately fivefold, than those in cluster II isolates (pi = 0.02001; Eta = 282) ($P <$ 0.05, $\chi^2$ test; $P < 0.05$, Fisher's exact test). Both clusters had negative Tajima's D values, while cluster I showed a significant value (Fig. 2D). Remarkably, the association between MLST cluster (I and II) and isolation source was insignificant, indicating a high degree of genotypic diversity among isolates from the same sources.

## The phylogenetic tree of *V. vulnificus* reveals the genomic divergence among the isolated sources

A phylogenetic tree was constructed to further understand the evolutionary relationship among the divergent isolated sources. This tree included diseased fish strains (VVA1, VVA6, JASA 1.6 APW, and CHOH1.7 BD) and compared them with 89 publicly available genomes from a wide range of geographical and ecological sources in the GenBank database based on single nucleotide polymorphisms (SNPs) (11). Among the 93 *V. vulnificus* genomes, the strains were separated into five clusters (Fig. 3). This result demonstrates that the four diseased fish isolates belong to cluster 1, including the majority group of clinical isolates (~61%), indicating that these four isolates seem to be potential foodborne pathogenic bacteria (Fig. 3). The distribution of *V. vulnificus* in cluster 2 showed striking divergence as the number of isolated sources was found at a comparable proportion. In contrast, cluster 3 and cluster 5 contained significant proportions of strains isolated from diseased fish (~46%) and humans (~90%), respectively ($P = 0.02$). The distinguishing distribution in cluster 4 is the relatively high proportion of both aquatic animal isolates (~41%) and clinical isolates (~59%).

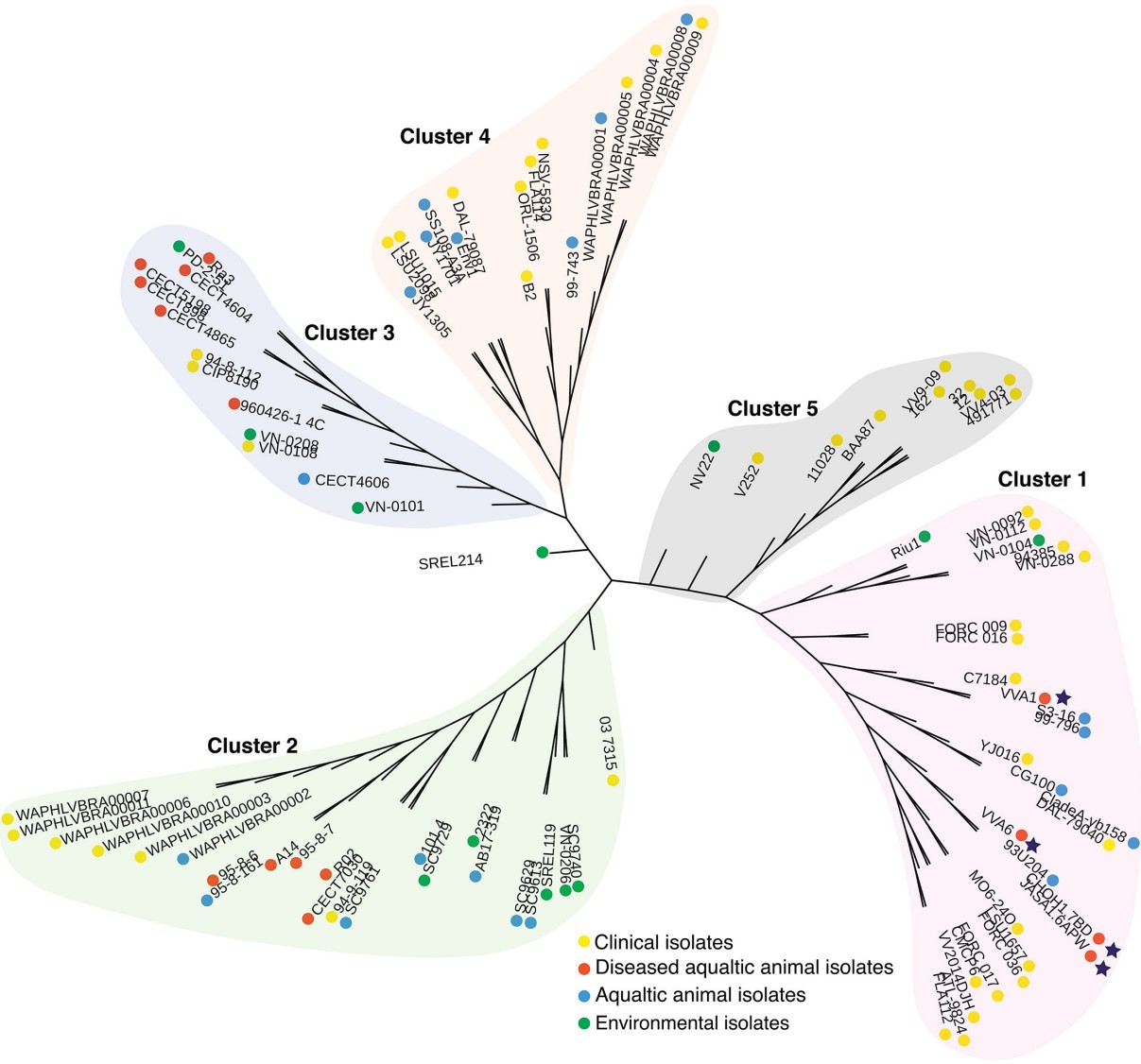

**FIG 3** Phylogenomic tree of *V. vulnificus* based on single nucleotide polymorphisms of the core genome using maximum likelihood.

## Genome analyses reveal genetic drivers associated with pathogenic potential strains

### Virulence, disease, and defense

In order to investigate the distribution of known virulence factors in diseased fish strains (VVA1, VVA6, JASA 1.6 APW, and CHOH1.7 BD), all coding regions were compared with those of CMCP6, YJ016, *Vibrio cholerae* O1 biovar El, *V. cholerae* O395, *Vibrio fischeri* ES114, and *Vibrio parahaemolyticus* RIMD 2210633 using Virulence Factors Database (41). The results showed that all diseased fish strains harbored similar putative virulence regions related to adherence, chemotaxis and motility, enzyme, iron uptake, and quorum sensing (Table S4). Interestingly, apart from those regions, the distribution of capsular polysac-charide clusters was diverse (Table S4). Only VVA1 possessed capsular polysaccharide-related genes (*rmlA* and *rmlC*) and type VI secretion-related genes (*vasA-K*), which were found in *V. fischeri* ES114 and in *V. cholerae* O1 biovar El and *V. cholerae* O395, respectively (Table S4).

## Cell wall and capsule

Capsular polysaccharide synthesis is an essential virulent factor of *V. vulnificus* to survive in human serum (34, 42), invade subcutaneous tissue (43, 44), and prevent opsonization/phagocytosis (45). We predicted proteins involved in the cell wall and capsule synthesis using Rapid Annotation using Subsystem Technology (RAST) version 2.0 (46). The results revealed that all diseased fish strains possessed similar protein clusters related to exopolysaccharide and *Vibrio* polysaccharide biosynthesis (Fig. 4A). Interestingly, three coding proteins related to sialic acid N-acetylneuraminic acid (Neu5Ac) and teichuronic acid biosynthesis were uniquely identified in the VVA6 genome (Fig. 4A). VVA6 also carried the N-acetylneuraminate cytidylyltransferase (*NeuA*), which involved

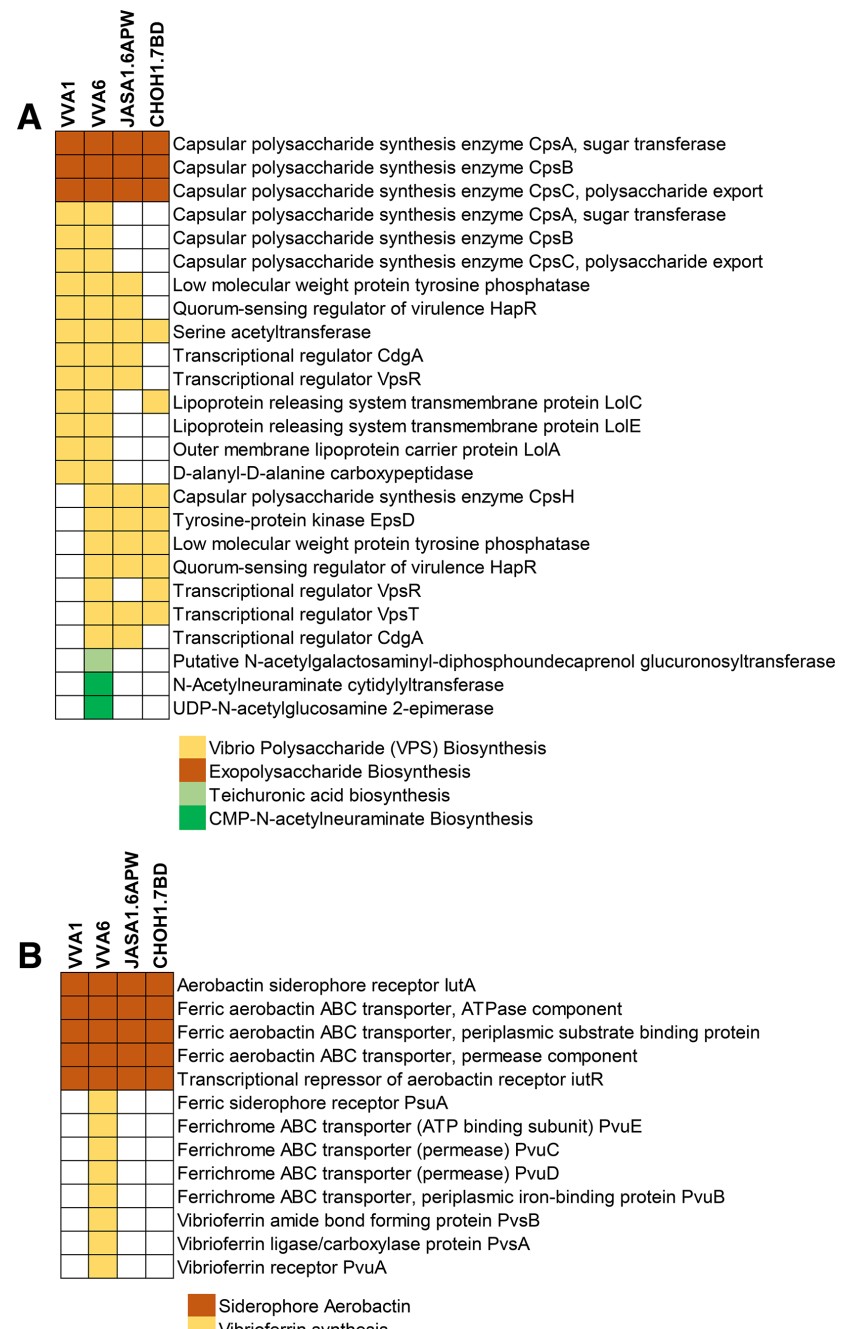

**FIG 4** Schematic representation of a comparison of (A) cell wall and capsule and (B) iron acquisition and metabolism found in diseased fish strains including VVA1, VVA6, JASA1.6APW, and CHOH1.7BD.

the final step of adding cytidine 5′-monophosphate to activate Neu5Ac (47). Modifying Neu5Ac has been employed by various bacteria, including *Neisseria meningitidis*, *Neisseria gonorrhoeae*, *Helicobacter pylori*, and *Salmonella enterica*, in order to evade the host immune response and enhance survival *in vivo* (48–52). Surprisingly, VVA6 also harbored putative N-acetylgalactosaminyl-diphosphoundecaprenol glucuronosyl-transferase responsible for teichuronic acid biosynthesis, which is usually found in cell wall component of Gram-positive (Fig. 4A) (53).

### Iron acquisition and metabolism

Almost all known *Vibrio* species produce siderophores as high-affinity iron-binding compounds. Siderophores are the most prevalent iron-scavenging mechanism in their natural habitats and within hosts (54). Alternative functions of siderophore have been proposed, including increasing virulence factor production (55) and protecting bacteria from oxidative stress created by antibiotics (56, 57). In this study, we investigated the coding proteins related to siderophore synthesis. The results showed that protein clusters involved in siderophore aerobactin could be detected in all diseased fish strains (Fig. 4B). Interestingly, only VVA6 carried a gene cluster responsible for vibrioferrin synthesis encoded by *pvuBCDE,* which could also be found in *V. parahaemolyticus* (Fig. 4B) (58–60).

### Distribution of prophage in V. vulnificus

To elucidate whether phage-like elements are integrated into *V. vulnificus* genomes, we detected those elements in each genome using PHASTER (61) and RAST version 2.0 (46). Interestingly, we found 160 phage-like proteins in *V. vulnificus* genomes, including 86 functional proteins (Table 1). The highest number of phage-like elements was identified in VVA6 compared to other strains. The majority of functional proteins are involved in their replication, DNA synthesis, and packing machinery (Table S5). Moreover, VVA1, JASA1.6APW, and CHOH1.7BD harbored one prophage integration site (attL and attR), while VVA6 carried two integration sites (Table 1).

### CRISPR arrays in V. vulnificus

The occurrence of CRISPR-Cas systems was previously found in 20% (106/528) of the family *Vibrionaceae* and in 11% (28/528) of *V. vulnificus* using the CRISPRCasFinder web server (62). As of the latest data from the CRISPR-Cas+++ database, five *V. vulnificus* strains harbored Cas cluster subtype I-E ($n = 1$), I-F ($n = 3$), and III-D ($n = 1$) (Table 2). The subtype I-F and I-E systems have been widely reported in *V. parahaemolyticus* and *V. cholerae* (63, 64). Our analysis of *cas* genes in four *V. vulnificus* genomes revealed that VVA1 and VVA6 harbored seven *cas* genes and two identical repeat arrays, classified as subtype I-E and III-D, respectively. In contrast, JASA1.6APW and CHOH1.7BD only carried the cas3 gene (Table 2).

### Diseased fish strain VVA6 demonstrates higher virulence to *Galleria mellonella* than VVA1

Based on the genome analysis, VVA6 appeared to be a more virulent strain than VVA1, as it possessed a higher number of potentially virulent-related genes. To determine their *in vivo* virulence, we used *G. mellonella* as an alternative infection model. The healthy larvae were injected with both strains at various bacterial concentrations ($10^2$–$10^7$ CFU/mL).

**TABLE 1**  Phage-like elements in *V. vulnificus* genomes

| *V. vulnificus* | CDS | Functional annotation | Hypothetical protein | attL | attR | tRNA |
|---|---|---|---|---|---|---|
| VVA1 | 21 | 15 | 6 | 1 | 1 | 1 |
| VVA6 | 55 | 31 | 24 | 2 | 2 | unclassified |
| JASA1.6APW | 42 | 20 | 22 | 1 | 1 | unclassified |
| CHOH1.7BA | 42 | 20 | 22 | 1 | 1 | unclassified |

**TABLE 2** The occurrence of CRISPR loci and Cas proteins in *V. vulnificus* genomes

| *V. vulnificus* | Origin | CRISPR type | Cas genes | Repeat consensus | Spacer count | GenBank number |
|---|---|---|---|---|---|---|
| VVA1 | Thailand | I-E | Cas1, Cas2, Cas5, Cas6, Cas7, Cse1, Cse2 | GTCTTCCCCACGCCCGTGGGGGTGTTTC | 21 | SAMN31632767 |
| | | | | GTCTTCCCCACGCCCGTGGGGGTGTTTC | 29 | |
| VVA6 | Thailand | III-D | Csm3, Csm3, Csm3, Cas10, Cas1, Cas2, Cas2 | GTTTCAGACATGCCCGGTTTAGACGGGATTAAGACT | 2 | SAMN31632768 |
| | | | | GTTTCAGACATGCCCGGTTTAGACGGGATTAAGACT | 6 | |
| CHOH1.7BA | Thailand | unclassified | cas3_TypeI, cas3_TypeI | GATATTTCTAACTGGGATACTTCCAATGTAAA | 3 | SAMN30916815 |
| JASA1.6APW | Thailand | unclassified | cas3_TypeI, cas3_TypeI | GATATTTCTAACTGGGATACTTCCAATGTAAA | 3 | SAMN30916812 |
| 07–2444 | Taiwan | I-E | Cas1, Cas2, Cas3, Cas5, Cas6, Cas7, Cse1, Cse2 | GAAACACCCCCACGTGCGTGGGGAAGAC | 73 | CP046835.1 |
| YJ016 | Taiwan | III-D | Cas10, Cas1, Cas2, Cas2, Csm3, Csm3, Csm3 | GTTTCAGACATGCCCGGTTTAGACGGGATTAAGAC | 2 | BA000038.2 |
| | | | | GTTTCAGACATGCCCGGTTTAGACGGGATTAAGAC | 9 | |
| 93U204 | Taiwan | I-F | Cas1, Cas3-Cas2, Cas6, Csy1, Csy2, Csy3 | GTTCACTGCCGTATAGGCAGCTTAGAAA | 55 | CP009261.1 |
| FORC_053 | South Korea | I-F | Cas1, Cas3, Cas2, Cas6, Csy1, Csy2, Csy3 | TTTCTAAGCTGCCTATACGGCAGTGAAC | 4 | CP015514.1 |
| FORC_036 | South Korea | I-F | Cas2, Cas3, Cas6, Csy3 | TCTTTAAGCCACCAGTGAGGTGGATAAC | 4 | CP015514.1 |

We then observed the larvae daily for 5 days for signs of the infection, including color changes from cream to black and slow movement. For larvae infected with VVA1 at $10^2$ and $10^3$ CFU/mL, all survived the 5-day period without gross signs of infection (Fig. 5A). The mortality rate for larvae infected with VVA1 at concentrations of $10^5$–$10^7$ CFU/mL began on day 1, with a survival rate of 3%–30%, dropping to 3%–10% by day 5 (Fig. 5A). In contrast, larvae infected with VVA6 showed black spots and slow movement after being challenged with $10^5$–$10^7$ CFU/mL on day 1. The survival rate was as low as 7% after receiving $10^2$ CFU/mL, and larvae did not survive after receiving $10^3$–$10^7$ CFU/mL (Fig. 5B). These results indicated that VVA6 demonstrated higher virulence in larvae than VVA1. This observation corresponded to the presence of putative coding protein related to virulence factors in VVA6. Therefore, the larval infection model may serve as a suitable alternative for assessing *V. vulnificus* virulence.

## DISCUSSION

The incidence of *V. vulnificus* infection has progressed rapidly with increasing climate warming, causing global deaths through either consumption of raw/undercooked

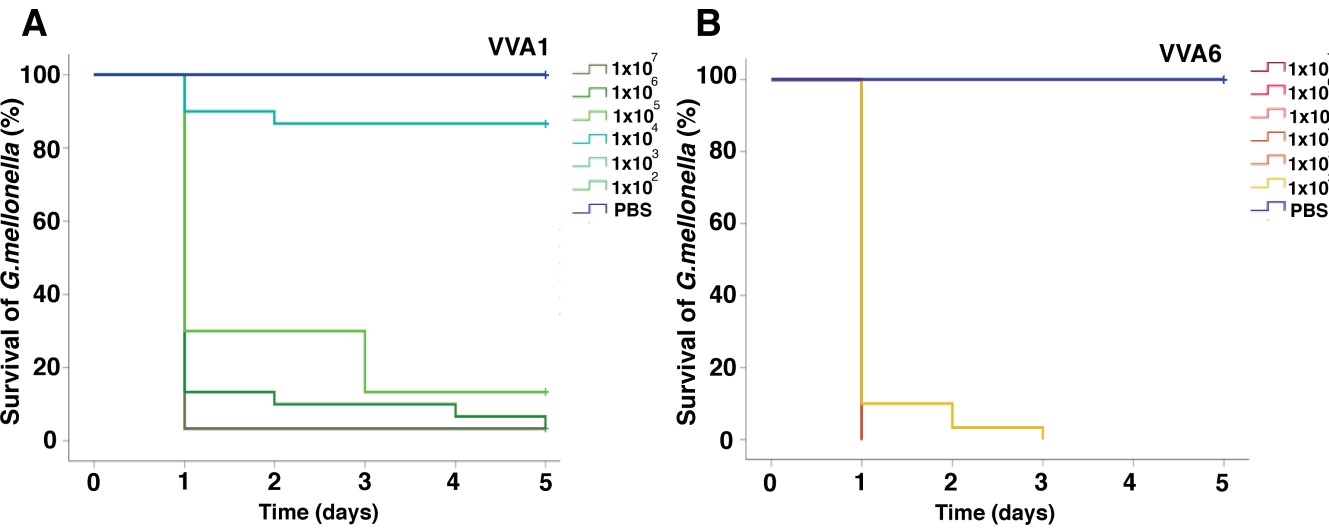

**FIG 5** Survival of *G. mellonella* following challenge with (A) VVA1 and (B) VVA6 with different bacterial concentrations.

seafood or direct contact (65–67). Apart from human infection, *V. vulnificus* is one of the critical zoonotic pathogens in aquatic animals, including brown-marbled grouper (39), crimson snapper (68), grass carp, and eels (69), leading to a substantial economic loss, particularly in tropical and temperate marine environments (70). Currently, the most significant challenge lies in specifying virulent *V. vulnificus* strains based on genetic markers. However, a high genetic diversity is observed between virulent and non-virulent lineages (71, 72). In this study, we aimed to distinguish virulent strains by investigating the distribution of genes or genetic regions related to virulence among *V. vulnificus* isolates from three different sources.

This study presented a high diversity of genotypes and phenotypes among the isolates, partly by source of origin (Fig. 1). Here, the *nanA* gene, responsible for sialic acid catabolism, was identified in all diseased fish and clinical isolates, and in approximately 65% of environmental isolates. *Vibrio* spp. capable of catabolizing sialic acid can outcompete non-catabolizers *in vivo,* enabling their survival and colonization in various tissues such as the jejunum, colon, respiratory, intestines, and urogenital tract (73, 74). Additionally, the PRXII was detected significantly more in diseased fish isolates (~84%) compared to environmental strains (~32%). The presence of both virulence-associated genes, *nanA* and PRXII, in environmental *V. vulnificus* strains suggests their potential as pathogens in both humans and aquatic animals (75).

However, the correlation analysis between the presence of other virulent genes/ regions and the molecular typing using MLVA and MLST indicated that virulent genes are distributed across various *V. vulnificus* clusters without a specific pattern confined to particular clusters. This finding challenges the use of these virulent genes/regions as definitive markers for risk assessment, contrasting with the previous understanding that these genes could reliably indicate virulence (75). Our result suggests a more complex scenario where these genes are present in a broader range of clusters than previously thought.

We then selected four diseased fish strains, CHOH 1.7 BD, JASA 1.6 APW, VVA1, and VVA6, which shared similar virulent regions (Fig. 1) for whole-genome sequencing to identify potential genetic and protein markers critical to virulence. Genomic comparison of these strains with representative strains in the database, using SNPS analysis, revealed that *V. vulnificus* strains from diseased fish were genetically associated with clinical strains, as evident in cluster 1 (Fig. 3). Despite the high genetic diversity among clusters, we could not establish a clear link between phylogenic patterns, isolation sources, and virulent capabilities.

Further investigation of known virulent factors in these bacterial genomes revealed divergent distributions of gene cassettes associated with capsular polysaccharides and iron acquisition systems. Notably, only VVA6 possessed the critical enzymes for synthesizing N-acetylneuraminic acid, a unique cell-surface structure modification that affects susceptibility to antibodies and phagocytosis (76, 77), thereby enhancing survival *in vivo* (48). The genomic and metabolic profiling of this carbohydrate in *Vibrionaceae* is highly diverse, suggesting various roles in environmental persistence and/or virulence (78–80).

Additionally, the VVA6 genomes harbored a gene cluster for vibrioferrin synthesis, a member of the carboxylate class of siderophores. The vibrioferrin originally reported in *V. parahaemolyticus* (58, 81, 82) has been less documented in *V. vulnificus*. The siderophore can be utilized by either the TonB1 system or the TtpC2-TonB2 system (83).

Apart from the presence of virulent regions in the genome, prophage elements were also identified in four diseased fish strains (Table 1; Table S5). Interestingly, no virulent gene was found within the prophage genomes. Additionally, the integrated mobile genetic elements, *attL* (left) and *attR* (right), detected in these strains, could facilitate horizontal transfer through conjugation and subsequent integration. The distribution of these prophage elements suggests their role in incorporating virulent genes, toxin genes, and metabolic pathways, thereby contributing to the genetic diversification and fitness advantages in *Vibrio* species such as *V. parahaemolyticus* (84), *V. cholerae*

(85), *Vibrio harveyi* (86), and *Vibrio anguillarum* (87). However, the functionality of these prophage elements, as predicted by *in silico* analyses, requires experimental validation. Induction experiments and cytotoxicity evaluations are necessary to understand their functional implications in virulence and environmental dissemination.

In addition to prophage elements, defense mechanisms, known as CRISPR–Cas systems, were observed in all diseased fish strains (Table 2) (88). VVA1 and VVA6 were classified into types I-E and III-D, respectively, while CHOH1.7BD and JASA1.6APW only carried the cas3 gene of type I. Notably, the presence of type I-E CRISPR-Cas has been associated with virulent factors, including thermostable direct hemolysin in *V. parahaemolyticus* clinical and food isolates (89). These systems are often linked with mobile genetic elements and are predominant in chromosomes of *Vibrio* spp. (~85%), contributing to their survival in harsh environmental conditions (90). The mechanisms by which prophage elements or CRISPR systems drive microbial evolution remain an area requiring further investigation.

We finally selected two *V. vulnificus* strains isolated from diseased fish, VVA1 and VVA6, to investigate their virulence against *G. mellonella*. This model is widely used as an alternative for studying the virulence of many bacteria, including *V. vulnificus* (91), with previous studies indicating a strong correlation between bacterial virulence in larvae and vertebrate models (92). In this study, VVA6 appeared to be more virulent than VVA1, correlating with the presence of associated virulent genes and genetic elements.

Our study contributes to the understanding that environmental strains of *V. vulnificus* may serve as reservoirs for virulent strains, suggesting a link between environmental factors and the emergence of virulence. While direct evidence on how these factors drive genetic divergence and evolution toward pathogenicity in humans and aquatic animals was not established, our findings indicate that environmental *V. vulnificus* strains could potentially become more virulent under certain conditions. This highlights the complexity of microbial evolution in natural habitats and its implications for public health. Acknowledging limitations such as small sample size, the lack of whole-genomic characterization of avirulent strains, geographic specificity, or methodological constraints that limited our scope of investigation, our study underscores the need for further comprehensive research. A comparative analysis of virulent and avirulent strains could provide invaluable insights into the evolution of pathogenicity from environmental reservoirs. Future investigations should aim to elucidate the intricate relationship between environmental conditions, genetic changes, and the development of virulence in *V. vulnificus*, providing deeper insights into the mechanisms that drive the transition of these bacteria from environmental strains to pathogenic ones.

## MATERIALS AND METHODS

### Bacterial strains and biotype determination

Eighty-five *V. vulnificus* strains from various sources and origins were used in this study, including *V. vulnificus* ATCC 27562, 41 environmental strains isolated from oyster farms in Surat Thani province, Thailand (2012), 10 clinical strains provided by the department of medical science, and 33 strains isolated from diseased fish by the department of fisheries, Songkhla, Thailand (2014–2015). Among these, 10 diseased fish isolates, part of the 33 mentioned, were previously identified as virulent in the tiger grouper model (5). Indole production was assayed to determine the biotype of *V. vulnificus*, with most categorized as biotype 1. Three strains, including two environmental isolates and one diseased fish isolate, were indole-negative and classified as biotype 2.

### Molecular analyses

The *vcg* type was identified in all isolates. Five genetic regions, including *manIIA*, *nanA*, nab region and PRXII were investigated using the method described in previous studies (29). Genomic DNA was extracted using the colony boiling method. Briefly, a full loop of bacteria was transferred to 100 µL deionized water, boiled at 100°C for 10 min, and

then placed on ice for 15 min. The PCR mixture contained 1× PCR buffer (0.25 mM dNTP, 1.5 mM MgCl$_2$), 1 µM of each forward and reverse primer, 1 U Taq polymerase (Bioline,UK), and 5 µL of DNA template in a total volume of 25 µL. PCR was performed in a Bio-Rad T100 Thermal cycler with the following conditions: initial denaturation at 94℃ for 1 min, followed by 30 cycles of 30 s at 94℃, 45 s at the melting temperature for each genetic region (Table S1), and 1 min at 72℃, with a final extension of 10 min at 72℃. PCR products were analyzed by agarose gel electrophoresis and visualized under UV light.

## Phenotype analyses

To assess mannitol fermentation in *V. vulnificus*, each isolate was inoculated into mannitol fermentation broth comprising 1% mannitol, 0.0075% bromothymol blue, 1% peptone, 0.5% NaCl (pH 7.4). The cultures were then incubated at 37℃ for 1 to 5 days. A change in culture broth color from blue to yellow indicated mannitol fermentation.

Hemolytic activity was tested according to the method described by Bier et al. (29) (29). Briefly, *V. vulnificus* was grown in brain heart infusion broth (BD Difco, France) supplemented with 1% NaCl at 37℃ and 150 rpm until reaching 1.6 optical density at 600 nm (late log phase). The cultures were then centrifuged at 6,000 × *g* and 4℃ for 10 min, and both supernatant and pellet were collected. The supernatant was filtered through a 0.22 µm membrane filter (Corning, Germany), and the pellet was resuspended in cold phosphate-buffered saline (PBS) (1 mM NaCl, 2.7 mM KCl, 10 mM Na$_2$HPO$_4$, 1.8 mM KH$_2$PO$_4$, pH 7.4) to a 1:10 dilution. To assess hemolytic activity, 500 µL of either filtered supernatant or the diluted pellet suspension were combined with an equal volume of 4% human red blood cells in PBS and incubated at 37℃ for 2 h. After centrifugation at 2,500 × *g* and 4℃ for 10 min, the released hemoglobin in the mixture was measured at OD 570 nm. For controls, 2% Triton X-100 in PBS and PBS alone were used as the positive and the negative control, respectively.

Serum susceptibility of all *V. vulnificus* strains was assessed using a colorimetric serum sensitivity assay (93). Initially, *V. vulnificus* isolates were cultured in 96-well plates containing Luria-Bertani broth (BD Difco, France) supplemented with 1% NaCl and incubated at 37℃ for 18 h. Subsequently, the culture was transferred to peptone glucose broth (1% glucose, 0.0075% bromothymol blue, 1% peptone, 0.5% NaCl, pH 7.4) with varying concentrations of human serum ranging from 0% to 80% (0, 10, 20, 40, 60, and 80%) and incubated at 37℃ for 24 h. A color change in the culture from blue to yellow indicated bacterial resistance to serum. The serum sensitivity of the isolates was classified into three groups: resistant (the ability to grow in 60 to 80% of human serum), intermediate (the ability to grow in 20 to 40% of human serum), and sensitive (the ability to grow in 0 to 10% of human serum).

## MLVA molecular typing

Twenty strains were selected based on their sources and origins, comprising five environmental strains, five clinical strains, five strains isolated from diseased fish, and five strains identified as fish pathogens. Eleven loci across chromosomes 1 and 2 (VV-0401, VV-1044, VV-1151, VV-2339, VV-2577, VV-2785, VV1-1615, VVA-0305, VVA-0375, VVA-0930, and VVA-1475) were investigated using a slightly modified method of (12, 13). Genomic DNA was prepared using the Presto Mini gDNA Bacteria kit, and the 11 loci were amplified by PCR (see Table S2). The PCR mixture contained 1× buffer (0.25 mM dNTP, 1.5 mM MgCl$_2$), 10 µM each of forward and reverse primers, 1 U Taq polymerase (Bioline, UK), and 50 ng DNA template in a total volume of 25 µL. PCR was conducted in a Bio-Rad T100 Thermal cycler (Bio-Rad, T100 Thermal Cycle, USA) with the following conditions: an initial denaturation at 95℃ for 5 min, followed by 35 cycles of 94℃ for 1 min, 55℃ for 30 s, and 72℃ for 1 min, with a final extension at 72℃ for 5 min. The amplified fragments were detected using a fluorescent dye (Table S2) in capillary electrophoresis (First BASE Laboratories Sdn Bhd, Malaysia). Repeat counts of each locus and phylogenetic patterns were analyzed and compared using GeneScan-500 LIZ as a standard, with data processed in Bionumeric version 7.6.

## Phylogenetic tree of housekeeping genes

Phylogenic analysis was performed on 10 housekeeping genes from 20 strains. The genes were amplified as described by Bisharat et al. (37) and subsequently sequenced by Novogene Co. Ltd., Singapore. Alignment of nucleotide sequences was conducted using ClustalX with default settings, and the alignments were manually refined using Geneious (version 2020.1.2). A maximum likelihood phylogenetic tree was then constructed, employing rapid bootstrapping with 100 replicates.

## Whole-genome sequencing and genome feature analysis

*V. vulnificus* genomic DNA was extracted using Presto Mini gDNA Bacteria kit following the manufacturer's instructions. DNA concentration was estimated using absorbance 260 with a Nanodrop spectrophotometer. DNA quality and integrity were assessed by measuring the absorbance 260/280 nm and performing agarose gel electrophoresis. Whole-genome sequencing was conducted using the MGISEQ-2000 system (BGI, Beijing, China) to generate 150 bp paired-end reads. Genomic sequences were *de novo* assembled with SPAdes Genome Assembler (94). Draft genomes were annotated using the NCBI Prokaryotic Genomes Automatic Annotation Pipeline and RAST online servers (46). For phylogenetic analysis, *V. vulnificus* sequences from NCBI public database were used. Core genome SNP analyses were performed using SNP sites (95), and the maximum likelihood trees were generated with Geneious (96) and visualized using Interactive Tree Of Life (97). Virulence factors, toxins, phage-like elements, and CRISPR type were identified using the Virulence Factor Database (41), PHASTER (61), and CRISPRCasFinder (62), respectively.

## Virulence of *V. vulnificus* in *G. mellonella*

*G. mellonella* larvae, obtained from a local vendor, were stored in the dark for 10 days before the start of the experiment. For each experiment, healthy larvae weighing approximately 250 mg (10 larvae) were selected. Bacterial cultures at log phase ($OD_{600}$ nm of 1.0) were centrifuged, and the pellet was washed twice with PBS. Bacterial suspensions were prepared in PBS at concentrations ranging from $10^2$ to $10^7$ CFU/mL, with PBS alone serving as a negative control. Ten microliters of each concentration was injected into the last left prolonged leg of the larvae. The larvae were kept in the dark at room temperature (approximately 25°C) for 5 days. Experiments were conducted in triplicate. Mortality and melanization in the larvae were monitored daily, and Kaplan-Meier survival curves were plotted.

## Statistical analyses

The relationships between genotype, phenotype characteristics, and stain source of *V. vulnificus* were analyzed using $\chi^2$ test and Fisher's exact test ($\alpha = 0.05$) in SPSS. The discriminative ability of these characteristics was quantified using Simpson's DI (98).

## ACKNOWLEDGMENTS

The authors would like to thank Songkhla Aquatic Animal Health Research Center for the fish disease *Vibrio vulnificus* isolates used in this study. In addition, this study was supported by the Division of Biological Science and Division of Computational Science, Faculty of Science, Prince of Songkla University for research facilities.

This work was supported by funding from the Government of Thailand (SCI560059S, SCI580246c, and SCI601141S).

A.N. was financially supported by the Postdoctoral Fellowship, Ratchadapisek Somphot Fund.

Conceptualization: A.N., R.P., Methodology: A.N., R.P., Investigation: A.N., K.S., J.M., Formal analysis: A.N., K.S., Visualization: A.N., K.S., R.P., Validation: A.N., K.S., K.W.J., R.P.,

Writing-Original Draft: A.N., R.P., Writing-Review & Editing: A.N., R.P., Project administration: R.P., Resources: J.T., R.P., Supervision: R.P., Funding acquisition: R.P.

## AUTHOR AFFILIATIONS

[1]Department of Biochemistry, Faculty of Science, Chulalongkorn University, Bangkok, Thailand

[2]Division of Biological Science, Faculty of Science, Prince of Songkla University, Hat Yai, Songkhla, Thailand

[3]Department of Biomedical Sciences and Biomedical Engineering, Faculty of Medicine, Prince of Songkla University, Songkhla, Thailand

[4]Translational Medicine Research Center, Faculty of Medicine, Prince of Songkla University, Songkhla, Thailand

[5]Division of Computational Science, Faculty of Science, Prince of Songkla University, Hat Yai, Songkhla, Thailand

[6]Department of Fisheries, Aquatic Animal Health Research and Development Division, Songkhla Aquatic Animal Health Research Center, Songkhla, Thailand

## AUTHOR ORCIDs

Ampapan Naknaen  http://orcid.org/0000-0002-5566-1207
Komwit Surachat  http://orcid.org/0000-0001-7793-7561
Jutamas Manit  http://orcid.org/0009-0008-1081-0054
Rattanaruji Pomwised  http://orcid.org/0000-0002-9541-3224

## FUNDING

| Funder | Grant(s) | Author(s) |
| --- | --- | --- |
| Government of Thailand | SCI6602051S | Rattanaruji Pomwised |
| Government of Thailand | SCI580246c | Rattanaruji Pomwised |
| Government of Thailand | SCI601141S | Rattanaruji Pomwised |

## ETHICS APPROVAL

The protocol of the experiment in the animal model has been approved by the Institutional Animal Care and Use Committee, Prince of Songkla University (MHESI 68014/621 and Ref. AQ023/2023).

## ADDITIONAL FILES

The following material is available online.

Supplemental Material

**Table S1 (Spectrum00079-24-s0001.docx).** Primers for *V. vulnificus* genotypic characterization.
**Table S2 (Spectrum00079-24-s0002.docx).** Primers for MLVA.
**Table S3 (Spectrum00079-24-s0003.docx).** Alleles detected across all 10 genes evaluated in *V. vulnificus* isolates in the MLST database.
**Table S4 (Spectrum00079-24-s0004.xlsx).** *V. vulnificus* strains with the Virulence Factors Database.
**Table S5 (Spectrum00079-24-s0005.xlsx).** Phage-like elements in each strain of *V. vulnificus* genomes.

## Open Peer Review

**PEER REVIEW HISTORY (review-history.pdf).** An accounting of the reviewer comments and feedback.

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
