## [Reviewer comments · Microbiology Spectrum]

Microbiology Spectrum

Virulent properties and genomic diversity of *Vibrio vulnificus* isolated from environment, human, diseased fish

Ampa Naknaen, Komwit Surachat, Jutamas Manit, Korakot Jetwanna, Jumroensri Thawonsuwan, and Rattana-ruji Pomwised

Corresponding Author(s): Rattana-ruji Pomwised, Prince of Songkla University

Review Timeline:

Submission Date:	January 18, 2024
Editorial Decision:	March 25, 2024
Revision Received:	April 7, 2024
Accepted:	May 2, 2024

Editor: Jing Han

Reviewer(s): Disclosure of reviewer identity is with reference to reviewer comments included in decision letter(s). The following individuals involved in review of your submission have agreed to reveal their identity: Xavier Greg Caguiat (Reviewer #1)

Transaction Report:

DOI: <https://doi.org/10.1128/spectrum.00079-24>

Re: Spectrum00079-24 (Virulent properties and genomic diversity of *Vibrio vulnificus* isolated from environment, human, diseased fish)

Dear Dr. Rattanaruji Pomwised:

Thank you for the privilege of reviewing your work. Below you will find my comments, instructions from the Spectrum editorial office, and the reviewer comments.

Revision Guidelines

Sincerely,
Jing Han
Editor
Microbiology Spectrum

Reviewer #1 (Comments for the Author):

Great work on *Vibrio vulnificus*. This is groundbreaking work in trying to compare molecular and biochemical aspect of the different strains from different sources.

Reviewer #2 (Comments for the Author):

The article complain the diversity among *V. vulnificus* isolates from different origins. Moreover, study and compare the presence of different genetic data among theses isolates. After read the manuscript I have some comments,

- 1) Table S4 and S5 were not include, please attach it.
- 2) Please add the statistics results. Only appear in Material and Methods but in results only the word significative are mentioned , but not the data.
- 3) Do you use negative control in the animals experiments?
- 4) Why use VVA1 and VV6 isolates? I think that could be useful compare some avirulent environment strains.
- 5) Why used *G. mellonella* instead of fish?
- 6) Line 269 please add the doc after N in N gonorrhoea.
- 7) Why you only select 10 strains to MLVA or MLST analysis? Why not all of them?

This article is easy to follow and the information is correlatable.

Dear Editor,

Thank you for your email and for forwarding the valuable comments from the reviewers regarding our manuscript entitled "**Virulent properties and genomic diversity of *Vibrio vulnificus* isolated from environment, human, diseased fish**" (ID: Spectrum00079-24). We greatly appreciate the reviewers' insights, which have been instrumental in enhancing the quality of our manuscript.

We have meticulously reviewed each comment and made corresponding revisions to address the concerns raised. These changes are marked in color in the marked-up version of the manuscript for easy identification. Below, we provide detailed responses to each of the reviewer's comments, indicating how we have addressed them in the revision.

Reviewer #1 (Comments for the Author):

Great work on *Vibrio vulnificus*. This is groundbreaking work in trying to compare molecular and biochemical aspects of the different strains from different sources.

Reply:

We thank the reviewer for the compliments and careful reading of the manuscript.

Reviewer #2 (Comments for the Author):

The article complain the diversity among *V. vulnificus* isolates from different origins. Moreover, study and compare the presence of different genetic data among theses isolates. After read the manuscript I have some comments,

1) Table S4 and S5 were not include, please attach it.

Reply:

We appreciate the reviewer's attention to detail in pointing out the omission of Tables S4 and S5. This oversight may have occurred during the transfer process between journals. We have now included both supplemental tables in the revised manuscript. Table S4 presents [*V. vulnificus* strains against the Virulence Factors Database], and Table S5 details [Phage-like elements in *V. vulnificus* genomes], both of which are crucial for supporting our findings and discussions in the text. We apologize for any inconvenience this may have caused and believe that the inclusion of these tables now strengthens our manuscript.

2) Please add the statistics results. Only appear in Material and Methods but in results only the word significative are mentioned , but not the data.

We apologize for the oversight in not providing detailed statistical results in the Results section. To address this, we have carefully reviewed the manuscript and added the specific statistical data that supports our findings. These additions include detailed P-values. The necessary revisions have been made in the manuscript at lines 153-156, 167-169, 172-174, and 243-244. We believe these changes now adequately reflect the statistical significance of our results and strengthen the manuscript. We thank the reviewer for highlighting this area for improvement.

3) Do you use negative control in the animals experiments?

Reply:

We apologize for the oversight in not previously specifying the use of a negative control in our animal experiments. Indeed, we used PBS buffer as the negative control, which was also used for preparing the bacterial suspensions. This control was critical for validating the experimental setup by ensuring that any observed effects could be attributed to the bacterial infection rather than the injection process or buffer solution. To improve clarity and completeness, we have now added this information to the manuscript in Line 503-505. We appreciate the reviewer's attention to this detail

4) Why use VVA1 and VV6 isolates? I think that could be useful compare some avirulent environment strains.

Reply:

We appreciate the reviewer's suggestion to compare virulent strains with avirulent environmental strains. The selection of VVA1 and VVA6 for our study was based on their previously demonstrated pathogenicity in fish models and the presence of specific virulent regions identified through preliminary genome sequencing. These strains were chosen to elucidate the virulence potential and genetic markers associated with pathogenicity in a well-defined system.

However, we acknowledge the value of including avirulent environmental strains for a more comprehensive comparison. The omission of these strains was primarily due to the scope of our current study focusing on the genetic and phenotypic traits of known virulent strains and the limitations in resources for whole genome sequencing of additional environmental strains without prior evidence of virulence markers.

Your comment has highlighted an important gap in our research, and we recognize the potential insights that comparing virulent and avirulent strains could offer. While our current

dataset does not include whole genome sequenced avirulent strains, we consider this an important direction for future research. Such studies could further illuminate the genetic basis of virulence in *V. vulnificus* and enhance our understanding of pathogenicity evolution from environmental reservoirs. We have now added a section discussing this limitation and the value of future comparative studies in the Discussion part of our manuscript (Lines 399-404)."

5) Why used *G. mellonella* instead of fish?

Reply:

We appreciate the reviewer's inquiry regarding our choice of *Galleria mellonella* over fish models for studying *V. vulnificus* virulence. Indeed, fish models have been pivotal in *V. vulnificus* pathogenicity studies. Specifically, both VVA1 and VVA6 strains were previously characterized in Tiger Grouper (*Epinephelus fuscoguttatus*) (see Reference 5). Nonetheless, *G. mellonella* offers an alternative, widely accepted model for bacterial virulence studies, providing significant ethical and logistical advantages. Studies have shown a strong correlation between bacterial virulence observed in *G. mellonella* and that in vertebrate models (see ref 92 and 93), underscoring its relevance and utility (line 387-392). Given considerations of financial support, reduced regulatory requirements, and our commitment to the principles of the 3Rs (Replacement, Reduction, and Refinement) in animal research, *G. mellonella* emerged as a pragmatic and ethically responsible choice for our virulence studies.

5. Thawonsuwan J, Kasornchandra J, Soonsan P, Keawtapee C. 2016. Isolation of *Vibrio vulnificus* Biotype I from Disease Outbreaks on Cultured Tiger Grouper *Epinephelus fuscoguttatus* Forsskal, 1775. *Fish Pathol* 51:S39–S45.

92. Yamamoto M, Kashimoto T, Yoshimura Y, Tachibana N, Kuroda S, Miki Y, Kitabayashi S, Tong P, Xiao J, Tanaka K, Hamamoto H, Sekimizu K, Yamamoto K. 2016. A silkworm infection model to investigate *Vibrio vulnificus* virulence genes. *Molecular Medicine Reports* 14:4243–4247.

93. Jander G, Rahme LG, Ausubel FM. 2000. Positive Correlation between Virulence of *Pseudomonas aeruginosa* Mutants in Mice and Insects. *J Bacteriol* 182:3843–3845.

6) Line 269 please add the doc after N in *N gonorrhoea*.

Reply:

We appreciate the reviewer's attention to detail and have corrected the typographical error by adding a dot after 'N' in '*N. gonorrhoeae*' as suggested. The revision can be found in Line 268-269.

7) Why you only select 10 strains to MLVA or MLST analysis? Why not all of them?

Reply:

We appreciate the reviewer's question regarding the selection criteria for MLVA and MLST analysis. Due to constraints on financial resources, we strategically selected 20 strains, ensuring a balanced representation across different isolation sources to capture genetic diversity within our study's scope. This selection was made to maximize the breadth of our analysis within the limitations of our funding.

Subsequently, based on the preliminary findings and to deepen our understanding of *Vibrio vulnificus* evolution, we prioritized 4 strains for whole genome sequencing. These strains were chosen for their distinct virulent regions, aiming to cover a broad spectrum of genetic variability. The SNP-based phylogenetic analysis, including publicly available sequences, was then employed to explore bacterial evolutionary patterns more comprehensively.

This approach was adopted to judiciously use our resources while still achieving meaningful insights into the genetic and evolutionary dynamics of *V. vulnificus*. We recognize the value of including a larger set of strains in MLVA and MLST analyses and consider this an important direction for future studies as more resources become available.

8) This article is easy to follow and the information is correlatable.

Reply:

We thank the reviewer for the compliments and careful reading of the manuscript.

We hope that our revisions and responses adequately address the reviewers' comments and that our manuscript is now suitable for publication in the Microbiology Spectrum. We are grateful for the opportunity to improve our work and look forward to any further feedback.

Best regards,

Dr. Rattanaruji Pomwised

Division of Biological Science, Faculty of Science
Prince of Songkla University, Hat Yai, Songkhla, Thailand
rattanaruji.p@psu.ac.th

Re: Spectrum00079-24R1 (Virulent properties and genomic diversity of *Vibrio vulnificus* isolated from environment, human, diseased fish)

Dear Dr. Rattanaruji Pomwised:

Your manuscript has been accepted, and I am forwarding it to the ASM production staff for publication. Your paper will first be checked to make sure all elements meet the technical requirements. ASM staff will contact you if anything needs to be revised before copyediting and production can begin. Otherwise, you will be notified when your proofs are ready to be viewed.

Sincerely,
Jing Han
Editor
Microbiology Spectrum

Reviewer #3 (Comments for the Author):

The article explores the diversity among *Vibrio vulnificus* isolates from various origins, focusing on the differences in their genetic makeup. It thoroughly examines and compares the genetic data among these isolates, emphasizing the variation in their molecular and biochemical traits. This research does an excellent job in its comparative analysis of the strains, significantly enhancing our understanding of *Vibrio vulnificus*.